# Magnetic origami creates high performance micro devices

Felix Gabler [1,2,5], Dmitriy D. Karnaushenko [1,5], Daniil Karnaushenko[1] & Oliver G. Schmidt[1,2,3,4]

Self-assembly of two-dimensional patterned nanomembranes into three-dimensional micro-architectures has been considered a powerful approach for parallel and scalable manufacturing of the next generation of micro-electronic devices. However, the formation pathway towards the final geometry into which two-dimensional nanomembranes can transform depends on many available degrees of freedom and is plagued by structural inaccuracies. Especially for high-aspect-ratio nanomembranes, the potential energy landscape gives way to a manifold of complex pathways towards misassembly. Therefore, the self-assembly yield and device quality remain low and cannot compete with state-of-the art technologies. Here we present an alternative approach for the assembly of high-aspect-ratio nanomembranes into microelectronic devices with unprecedented control by remotely programming their assembly behavior under the influence of external magnetic fields. This form of magnetic Origami creates micro energy storage devices with excellent performance and high yield unleashing the full potential of magnetic field assisted assembly for on-chip manufacturing processes.

[1] Institute for Integrative Nanosciences, Leibniz IFW Dresden, 01069 Dresden, Germany. [2] Material Systems for Nanoelectronics, TU Chemnitz, 09107 Chemnitz, Germany. [3] Research Center for Materials, Architectures and Integration of Nanomembranes (MAIN), TU Chemnitz, 09126 Chemnitz, Germany. [4] Nanophysics, Faculty of Physics, TU Dresden, 01062 Dresden, Germany. [5] These authors contributed equally: Felix Gabler, Dmitriy D. Karnaushenko. Correspondence and requests for materials should be addressed to D.K. (email: d.karnaushenko@ifw-dresden.de) or to O.G.S. (email: o.schmidt@ifw-dresden.de)

Since the late 1960s[1], self-assembly is known to play a crucial role in the formation of three-dimensional (3D) biological structures and has become increasingly appealing in research laboratories all over the world to construct complex 3D architectures and electronic devices on the nanoscopic and microscopic scale[2–4]. It is, however, striking that self-assembly of mass-produced 3D electronic devices has failed to enter any industrial-level manufacturing schemes even nowadays. Among various self-assembly strategies, so-called "micro-Origami" which is the art of self-folding two-dimensional (2D) nanomembranes into micro-architectures[4–9], has opened up novel ways in constructing 3D mesoscale devices of "Swiss-roll"[10–16], polyhedral[17,18] and even more complex[19,20] shapes benefiting from state-of-the-art wafer-scale manufacturing technologies[21–23]. However, mesoscale 3D self-assembly of nanomembrane devices faces severe challenges associated with low yield, wide parameter spreading, and bad reproducibility.[24,25]

The rigorous miniaturization of micro-electronic devices requires equally resolute advancement in the development of micro-energy storage technologies.[26–28] Among many different forms of micro-energy storage devices, electrostatic and electrolytic capacitors are by far the most demanded and abundant components in any electronic circuit.[29] The large number of these components becomes increasingly critical for size and weight limits of the overall system where each of such an element must provide high-performance electrical characteristics measured per volume or footprint area.[30] Recently, strain-driven 3D self-assembly of nanomembranes[31] has been successfully applied in manufacturing compact "Swiss-roll" electrostatic capacitors[10,32,33] promising high volumetric efficiency and capacitance densities obtained via a parallel and scalable fabrication process. In order to improve the performance of 3D energy storage devices, it is essential to use high length-to-width aspect ratio nanomembranes. Reliable and reproducible self-assembly of such nanomembranes has become, however, an increasingly challenging task as tiny fluctuations in the formation process lead to large deviations in the final shape and structure of each device. From a fundamental point of view, these deviations are triggered and amplified by a large number of spatial degrees of freedom (DoF) leading to various pathways through which nanomembranes relax available potential energy during their assembly process. The assembly pathways are defined by complex potential energy landscapes (PELs), similar to those responsible for the structure and dynamics of atomic, mesoscopic, and macroscopic systems in general[34–38]. It is crucial not only to understand the shape of the PEL analysing their features such as local minima, energy barriers, and slopes[39] but also to modify the PEL on demand via, for instance, adjusting the system's DoF.

Here we overcome these issues by introducing an alternative approach that explores the magnetic field-assisted assembly of 3D nanomembrane architectures. By applying static and dynamic magnetic fields during assembly, we create high-performance energy storage devices and demonstrate that cleverly guided micro-Origami can make a crucial difference when it comes to exploring viable technology routes towards exciting new manufacturing scenarios.

## Results

**Magnetic field-assisted assembly of 3D nanomembranes**. The self-assembly of a nanomembrane into a "Swiss-roll" architecture (SRA) is depicted in Fig. 1a. The nanomembrane in its initial planar state possesses a high aspect ratio between its length and width. We identify four main DoF (Fig. 1b) during the assembly process. The SRA can rotate around the $y$ (TILT) and $z$ (ROLL) axes, while rotation around the $x$-axis is prohibited because of the

constraints exerted by the substrate on the architecture. Windings of this architecture can also slide against each other, which is defined as a SHIFT of the structure during the assembly process. In this work, we consider a constant shift of all windings in one or another direction along the $z$-axis. Both $y$ (TILT) and SHIFT are undesirable DoF that lead to misassembly of the structure and as a result to lower yield of the SRA.

We incorporate thin magnetic films into the namomembrane and superimpose a magnetic field $\vec{H}$ during the assembly process to effectively reduce one of the DoF. We consider two distinct cases where the magnetic field either is oriented statically along the SRA axis (Fig. 1c) or rotates dynamically around the SRA axis (Fig. 1d). In these cases, either the $y$ (TILT) or the $z$ (ROLL) rotational DoF can be reduced, respectively. Torques, exerted by the external field $\vec{H}$ on the induced magnetic dipoles, align the SRA along the field, similar to a compass needle aligned by the magnetic dipole of the Earth. The DoF associated with SHIFT, however, remains present in both cases. In the case of the static axial magnetic field, the ferromagnetic layers magnetize accordingly and a torque $\vec{\tau_y}$ will be exerted onto the structure aligning it with the external magnetic field. This can be considered as elimination of $y$ (TILT) during the self-assembly process (Further details can be found in Supplementary Note 1.). However, SHIFT is still present but can be suppressed by patterning the initial planar structure into a trapezoid (Fig. 1a). The width of the structure is uniformly decreasing along the assembly pathway defined as a trapezoidal pitch. This initial shape of the nanomembrane leads to an offset between the edges of windings assembling into the SRA. Once assembled into the SRA, the ferromagnetic windings experience axial dipolar repelling forces at their edges (Fig. 1c), which keeps them in place and result in a suppressed SHIFT with only the $z$ (ROLL) available for the strain-driven self-assembly of the mesoscopic architecture.

In the case of an external magnetic field oriented radially (Fig. 1d), the $z$ (ROLL) should be effectively pinned to the external radial $\vec{H}$ magnetic field, which, if rotated around the $z$-axis of the SRA produces an axial torque $\vec{\tau_Z}$, similar to the torque exerted by a rotor in an electromotor. In this configuration, the self-assembly is required only to form the few first windings of the SRA, which we define as the rotor (Fig. 1d). The rotor carries radially aligned ferromagnetic structures as shown in Fig. 1d. In this approach, the remaining part of the "Swiss-roll" can be assembled by rotating the external radial magnetic field around the $z$-axis (Fig. 1d) realizing an efficient mesoscale tool for parallel assembly of cylindrical structures. However, the rotating magnetic field may still exert a torque $\vec{\tau_y}$ inducing $y$ (TILT) in the whole assembly. This tilt can be cancelled by choosing an appropriate geometry (width and sector) of the ferromagnetic structures inside the rotor. In the following, we analyse our results in terms of a PEL projected into free energy surfaces (FES) and torque surfaces (TS) associated with different assembly states and geometric parameters of the magnetic structures, which were obtained via finite element (FEM) simulations (ANSYS package). Furthermore, we design, fabricate, and assemble "Swiss-roll" parallel plate capacitors carefully recording fabrication yield and quality of the geometry for each orientation of the $\vec{H}$ magnetic field.

**Assembly in an axial magnetic field**. Prior to design and fabrication, we simulated the PEL of a "Swiss-roll" accounting for $y$ (TILT), SHIFT, and $z$ (ROLL). All other parameters such as thickness and material properties were fixed and can be found in the "Methods" section. The simulations were performed for rectangular and trapezoidal nanomembrane patterns adiabatically

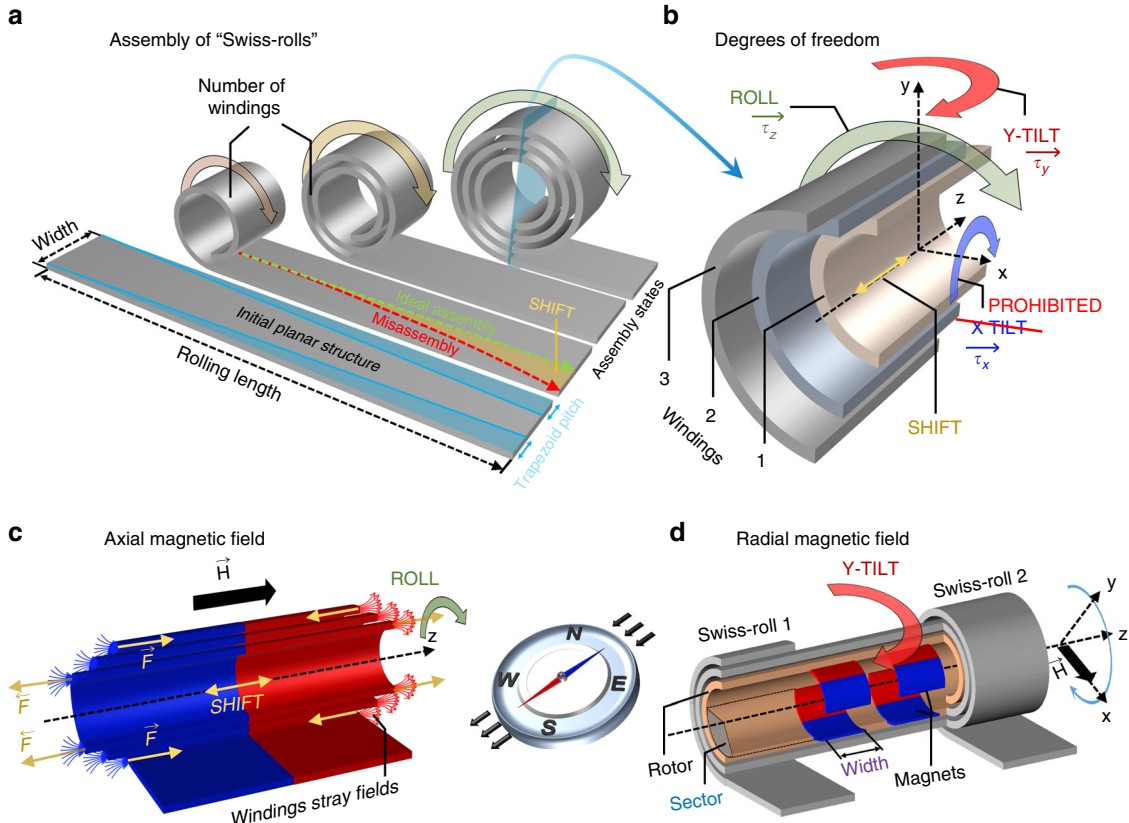

**Fig. 1** Schematics of the "Swiss-roll" assembly and possible degrees of freedom. **a** "Swiss-roll" assembly process that starts with an initially planar structure of a specific shape, either rectangular or trapezoidal with a defined pitch. **b** Degrees of freedom available for "Swiss-roll" architecture during the assembly process. **c** Axial magnetic field-assisted self-assembly of a "Swiss-roll" that is performed from a trapezoidal initial shape where magnetostatic forces play a key role in stabilization of the assembly. **d** Radial magnetic field-assisted assembly of a "Swiss-roll" where rolling of the nanomembrane is achieved with a self-assembled rotor in a rotating external magnetic field

assembling into SRAs as depicted in Fig. 1a. In this way, we obtain a value of the potential energy for each point of the PEL coordinate system, ranging from $-15$ to $+15\,\mu m$ for SHIFT, up to three windings for $z$ (ROLL), and a range of angles for $y$ (TILT). Two types of FES can be constructed from the simulations: one for magnetostatic and the other for mechanical interactions. The mechanical FES show a constant downslope of 50 pJ/winding (see Supplementary Fig. 1 and discussion in Supplementary Note 1) with no substantial difference between rectangular and trapezoidal design for the $y$ (TILT) DoF. This slope directly represents the driving force for the self-assembly by releasing the intrinsic strain and rolling around the $z$-axis, which causes propagation of the SRA along the $x$-axis (Fig. 1a, b). However, the magnetostatic FES (Figs. 2 and 3) plotted for the SHIFT DoF reveal substantially different shapes providing deep insight into the assembly behaviour for rectangular and trapezoidal nanomembrane SRAs.

Figure 2a shows the magnetostatic energy for three completed windings where the energy value at zero SHIFT is defined as zero. For the rectangular design, the desired "ideal" position of the windings (zero SHIFT) is located at the local maximum of the energy (Fig. 2a black). However, the system is very likely to minimize its potential energy by shifting the windings away from the centre into either the positive or the negative direction following a favourable pathway across the energy landscape. As the system always tends to minimize its potential energy, the probability of the system is high to slide left or right down the slope of the FES causing unwanted misassembly.

The trapezoidal design, in contrast, forms an energy plateau around zero SHIFT. Additionally, two new energy maxima appear and confine the plateau within $\pm Nt$, where $N$ represents the number of windings and $t$ the trapezoidal pitch (defined in $\mu m$/winding). The specific shape of the energy plateau can be influenced by the value of the trapezoidal pitch. For a trapezoidal pitch of 1 $\mu m$/winding (Fig. 2a blue), the energy plateau has a concave shape with a local minimum at zero SHIFT. Within this concave energy plateau, the system minimizes its potential energy by keeping the windings in the center of the plateau, which results in stabilization of the SRA. For trapezoidal pitches >1 $\mu m$/winding, for example, 2 $\mu m$/winding, the energy plateau becomes convexly shaped and two local minima on either side of the plateau can be seen along the SHIFT axis (Fig. 2a red, also see discussion in Supplementary Note 2 and Supplementary Fig. 2). Comparing the pitches, it is clear that the trapezoidal structure with a pitch of 1 $\mu m$/winding represents an optimal geometry for the initial planar structure. The impact of the planar nanomembrane shape on the assembly of the SRA is sketched in Fig. 2b, c. Sections of two adjacent windings with different states of SHIFT between them for both rectangular and trapezoidal design are depicted. For the rectangular case, magnetostatic interactions at the edges of the nanomembrane produce unidirectional axial forces (Fig. 2b) that push the edges away from each other and, as a result, misalign the windings. The sum of these forces is given in black in Fig. 2d. In contrast, magnetostatic interactions at the edges of the trapezoidal nanomembrane produce counteracting forces (Fig. 2c), which compensate each other when the windings are aligned in the middle of the SRA (Fig. 2d blue and red). Either

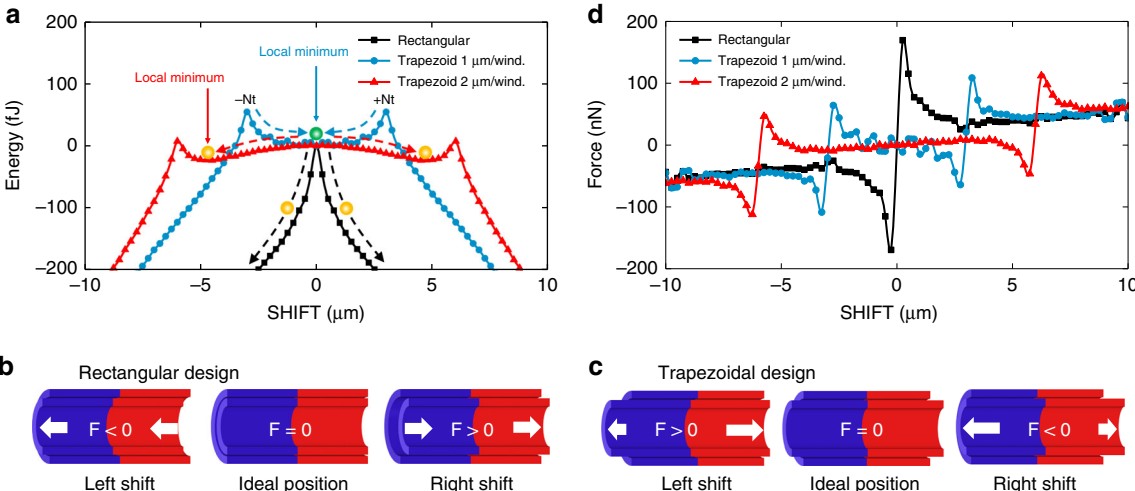

**Fig. 2** Influence of nanomembrane planar design on energy and force during assembly. **a** Magnetostatic energy as a function of shift for the rectangular design and the trapezoidal design with different pitch at three windings. Change from rectangular to trapezoidal pattern results in the formation of energy plateaus with new local minima. The specific shape of energy plateaus and the position of minima can be influenced by the trapezoid pitch. **b** Direction of axial forces $F$ depending on shift for two adjacent windings in the rectangular design. **c** Same as in **b** but for trapezoidal design. **d** Forces as a function of shift for the rectangular design and the trapezoidal design with different pitch at three windings

**Fig. 3** Shape-dependent behaviour of magnetic nanomembrane. **a** Magnetostatic free energy surfaces (FES) of rectangular design and corresponding experimental result of the device assembly. **b** Magnetostatic FES of trapezoidal design with a pitch of 1 μm/winding and corresponding experimental result of the device assembly. **c** Cross-sections of the FES in **a** with increasing number of windings. **d** Cross-sections of the FES in **b** with increasing number of windings. Scale bars are 1 mm

of these forces reaches its maximum when the left or right edges of the windings almost align due to the SHIFT. If the misalignment is larger (one of the winding edges is shifted beyond the edge of the other winding, i.e., $|\text{SHIFT}| > Nt$), one of the forces suddenly flips orientation and the winding is pushed out of the SRA following the downslope in the FES diagram similar to the rectangular case.

The entire 3D magnetostatic FES of the rectangular and trapezoidal design with an optimal pitch of 1 µm/winding are shown in Fig. 3. Green arrows at zero SHIFT denote the ideal assembly pathway of the SRA across the FES. Along this line, there is an upslope of 0.5 pJ/winding increasing the magnetostatic potential energy, which is, however, much smaller compared to the mechanical downslope (see Supplementary Fig. 1c). Having no substantial influence on the assembly process, the magnetostatic slope at zero SHIFT is subtracted from the FES. A closer look at the evolution of the FES cross-section with increasing number of windings (Fig. 3c, d) shows no difference between the two designs until a full winding has formed. Only further rolling into the second and third winding produces a difference in energy profiles that should stabilize and align the windings near the axial center of the SRA assisted by magnetostatic forces. This emphasizes the importance of selecting an appropriate trapezoidal pitch in order to accommodate the majority of statistically relevant SHIFT values within $\pm Nt$ that can occur during the formation of the first two windings as well as to avoid the formation of a convex-shaped energy plateau as described before. A trapezoidal design with an appropriate pitch is therefore expected to assemble into an SRA with much smaller or no SHIFT compared to a rectangular design, which we confirm experimentally and discuss in the following section.

**Design and verification in an axial magnetic field.** We designed and fabricated an array of double plate capacitor structures on Si/SiO$_2$ substrate samples $14 \times 14$ mm$^2$ in size. The samples accommodate both rectangular and trapezoidal (pitch 1 µm/winding) nanomembranes of the initially planar structure prepared in a parallel wafer-scale process. These structures are capable of self-assembled roll-up, which is driven by a built-in strain gradient induced during deposition of the nanomembrane materials. Fabrication steps are schematically presented in Fig. 4a, which begin with 2 nm of methyl cellulose (MC) deposited by spin coating on top of the substrate. This layer is soluble in water and acts as a sacrificial layer (SL) to release the pre-strained layer stack[40]. The SL is covered with 7 nm of aluminium oxide (Al$_2$O$_3$) deposited by atomic layer deposition (ALD) to protect the SL from humidity and water-based chemicals. Next, the capacitor stack is composed of 7 nm ALD Al$_2$O$_3$ sandwiched between two strained electrodes containing 15 nm of Ni and 20 nm of Cr deposited by electron-beam physical vapour deposition (eB-PVD). The strain in the layer stack can be controlled by deposition parameters as, for example, temperature, pressure, and deposition rate[41] that are specified in the "Methods" section. Contact areas are opened by reactive-ion etching (RIE) of Al$_2$O$_3$ with a subsequent deposition of Cr/Au pads by eB-PVD. Finally, windows (Fig. 4b) in Al$_2$O$_3$ are opened by RIE to expose the SL at the desired start edge. Assembly of the capacitors is accomplished by placing the substrates into a homogeneous magnetic field[42] of 796 kA/m (10 kOe) and applying water to initiate the release of the pre-strained layer starting at the opened windows (Fig. 4c). Driven by strain relaxation, the nanomembranes self-assemble into compact 3D SRAs (Fig. 4c inset and 4d) while tearing off the Al$_2$O$_3$ layer along the left and right edges of the structures.

Images acquired after the assembly process (Fig. 3a, b bottom) confirm the predicted behaviour of both designs. The rectangular design results in misassembly for all structures on the sample, while the trapezoidal design demonstrates ideal assembly with a total SHIFT <25 µm after about 50 windings for >90% of the structures (For more details, see Supplementary Note 3 and Supplementary Fig. 3a.). During assembly, the speed of "Swiss-rolls" moving along the $x$ axis is constant (no acceleration or deceleration) implying that all of the forces in the system are compensated, i.e., the rest of the intrinsic strain energy sinks into tearing the Al$_2$O$_3$ layer along the edges. This behaviour was confirmed by thinning the Al$_2$O$_3$, which leads to a higher rolling speed. The speed is four orders of magnitude higher than the assembly speed in previous reports[10,33] and can be in the range of 100 to 500 µm/s depending on the built-in strain, nanomembrane width and thickness of Al$_2$O$_3$ along the edges. The initially planar capacitor structures with a length of 10 mm and a width of 300 µm assemble into compact SRAs with 60 µm outer diameter reducing their footprint by more than two orders of magnitude. This effect leads to a footprint area of only 0.018 mm$^2$, which is smaller than the smallest industrial surface mounting device (SMD) capacitor (EIA 008004 [Footprint: 0.031 mm$^2$]). The Swiss-roll capacitor weighs 2 µg, which is almost an order of magnitude less than the smallest commercial device (around 16 µg for GRM011R60J103KE01L Murata Manufacturing Co., Ltd.). The electrical plot (Fig. 4e) demonstrates high-performance capacitive behaviour up to 2 MHz with a phase angle close to −90° <1 kHz. The planar structure has a 16-nF capacitance, which corresponds to a footprint areal capacitance density of 5.3 nF/mm$^2$. The assembly process adds extra capacitance as the bottom oxide layer contacts (Supplementary Fig. 3b) the top metal layer forming a second capacitor in parallel. Thus the 27-nF capacitance of the "Swiss-roll" devices is 70% higher than that of their planar counterparts while occupying <1% of its original footprint area. This translates into an industry-grade[30] areal capacitance density for electrostatic capacitors of 1500 nF/mm$^2$, which is five times higher compared to industry's smallest SMD capacitor capacitance density of 320 nF/mm$^2$ (EIA 008004, 10 nF, GRM011R60J103KE01L, Murata Manufacturing Co., Ltd.). We also fabricated capacitors with 15 nm of Al$_2$O$_3$ dielectric showing 9 nF planar capacitance (3 nF/mm$^2$) and 16 nF (889 nF/mm$^2$) after the self-assembly process. The dielectric thickness is, however, a compromise between high capacitance and high breakdown voltage. For 7 nm dielectric, the breakdown voltage is about 1.8 V, while the capacitors with 15 nm dielectric thickness can be operated at 6 V with leakage currents of few nA (Supplementary Fig. 3c) revealing an unprecedented high (>10$^4$ C/m$^3$) volumetric efficiency among any electrostatic capacitors available to date. The performance of the capacitors might be even further enhanced by including 2D, graphene layers and other all carbon materials into the layer stack[43,44]. All the details on the developed set-up (Supplementary Fig. 4a) for axial magnetic field-assisted self-assembly of nanomembranes, magnetic field uniformity (Supplementary Fig. 4b), and directionality (Supplementary Fig. 4c) are given in the Supplementary Note 4.

**Assembly in rotating radial magnetic field.** A critical degree of freedom (Fig. 1d) is the tilting due to the torque $\overrightarrow{\tau_y}$ around the $y$ axis that reduces the quality of the assembly or even damages the structure. Unlike the previous case, here the torque $\overrightarrow{\tau_x}$ cannot be ignored anymore because the planar nanomembrane is entirely freestanding and the orientations of the magnetic structures change with respect to the substrate following the rotation of the magnetic field vector $\overrightarrow{H}$ around the $z$ axis. For correct assembly, these torques should be minimized or engineered in such a way that the tube axis is oriented perpendicular to the external field. The value of the useful torque $\overrightarrow{\tau_Z}$ should be, however, kept at a

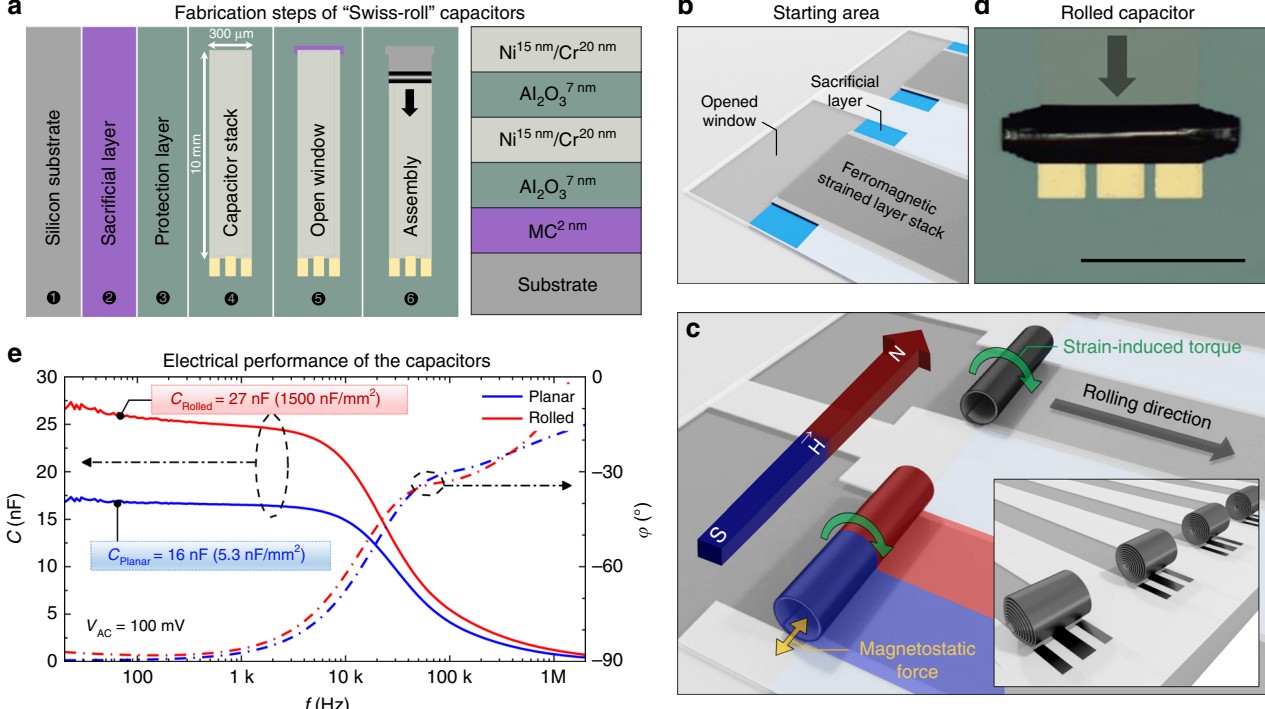

**Fig. 4** Fabrication of electrostatic capacitors using axial static magnetic field assembly. **a** Capacitor fabrication steps and cross-section of planar layer stack. **b** Rolling process is initiated by wet release of the ferromagnetic-strained layer stack from the sacrificial layer starting at the opened windows. **c** During the strain-induced self-assembly, the capacitors are stabilized by an axially (perpendicular to desired rolling direction) applied magnetic field $\vec{H}$ and the arising magnetostatic forces between the magnetized windings. The inset illustrates the final assembly state. **d** Optical microscopic image of the assembled capacitor. Scale bar is 200 μm. **e** Capacitance and phase angle as a function of frequency including areal capacitance density for planar and rolled state

maximum in order to be able to curl the nanomembrane over a long distance overcoming the continuous increase of mechanical potential energy due to the nanomembrane's mechanical restoring force.

We calculated the $\vec{\tau_x}$, $\vec{\tau_y}$, and $\vec{\tau_Z}$ TS in Fig. 5a for the geometric parameters of the magnetic stripes (Fig. 1d), namely, the width and the sector. To find the torques, we assumed a rotating magnetic field strength of 80 kA/m (1 kOe) and a soft ferromagnetic material (Co$_{90}$Fe$_{10}$ with coercivity of 4 kA/m or 50 Oe magnetized in the same direction as field $\vec{H}$) calculated in a spherical coordinate system with angles $\varphi$ and $\theta$ as shown in Fig. 5a–I. We set $\varphi = 90°$ and $\theta = 45°$ as a natural orientation of the structure with respect to the field for the assembly process. Under these assumptions, Fig. 5a displays an isotherm with the desired zero torque value for $\vec{\tau_y}$. However, along this isotherm there is no set of geometric parameters for which $\vec{\tau_x}$ can be cancelled without cancelling $\vec{\tau_Z}$, which would lead to a highly probable misassembly of the SRA.

In order to gain deeper insight into this issue, we calculated values of potential energies for each orientation ($-180° < \varphi < 180°$ and $-180° < \theta < 180°$) of the SRA with respect to the external constant magnetic field and plotted the FES for three different magnetic stripe geometries in Fig. 5b–d. These FES reveal various periodic landscapes and one common feature, which is associated with a rotational ambiguity around the magnetic field direction. In the given spherical coordinates system, this leads to, for instance, a valley minimum with a constant value of energy along the $\varphi$ axis observed in these FES at $|\theta| = 0°$; 180° etc. However, this extreme case is never reached because of the restoring force and the associated mechanical potential energy of the nanomembrane. This mechanical potential energy increases during the assembly process with a rate of 151 pJ/winding (see further details

in Supplementary Note 5 and Supplementary Fig. 5) for a 300-μm wide nanomembrane similar to a clock spring that should be compensated by the potential energy (depicted as arrows in Fig. 5b–d) of the rotor in the rotating external magnetic field. Green arrows indicate ideal pathways with increasing magnetic potential energy when the SRA is correctly orientated ($|\varphi| = 90°$) with respect to the external magnetic field. However, for an initial arbitrary shape of magnetic stripes (sector = 153° and width = 31 μm), which is illustrated in Fig. 5a–I, the FES is symmetric and possesses a convex shape as shown in Fig. 5b. This shape of the magnetic FES forces the rotor to relax (green–red arrows) into one of the local minima (e.g., located at $|\varphi| = 0°$; 180°) rotating the axis of the structure towards the undesired direction along the magnetic field and leading to misassembly of the SRA.

The TS (Fig. 5a) exhibits a point (sector = 147°; width = 16 μm, shown in Fig. 5a–II) located on the $\vec{\tau_y} = 0$ nN/m isotherm, where $\vec{\tau_Z}$ and $\vec{\tau_x}$ are equivalent and show their highest, about 2.9 nN/m, $\vec{\tau_Z}$ torque for this isotherm. Despite the non-zero value of $\vec{\tau_x}$, the corresponding FES (Fig. 5c) possesses three characteristic concave energy profiles located at $|\varphi| = 0°$; 90°, where the SRA structure predominately relaxes during the assembly process. Between these concave profiles, there are characteristic energy barriers with a relative height of about 1 nJ that prevent transitions from one pathway to the other and as a result stabilizing the $\varphi$ angle at one of these values, where $|\varphi| = 0°$ is undesired and indicated as the white arrow in Fig. 5c. However, we are free to choose the initial direction of the magnetic field to be $|\varphi| = 90°$ rendering this design stable for the magnetic field-assisted assembly as indicated by green arrows in Fig. 5c. We plotted another FES (Fig. 5d) for a different point on the TS (width = 2.5 μm; sector = 164°) demonstrating two similar concave profiles at the desired $|\varphi| = 90°$ coordinates favouring two

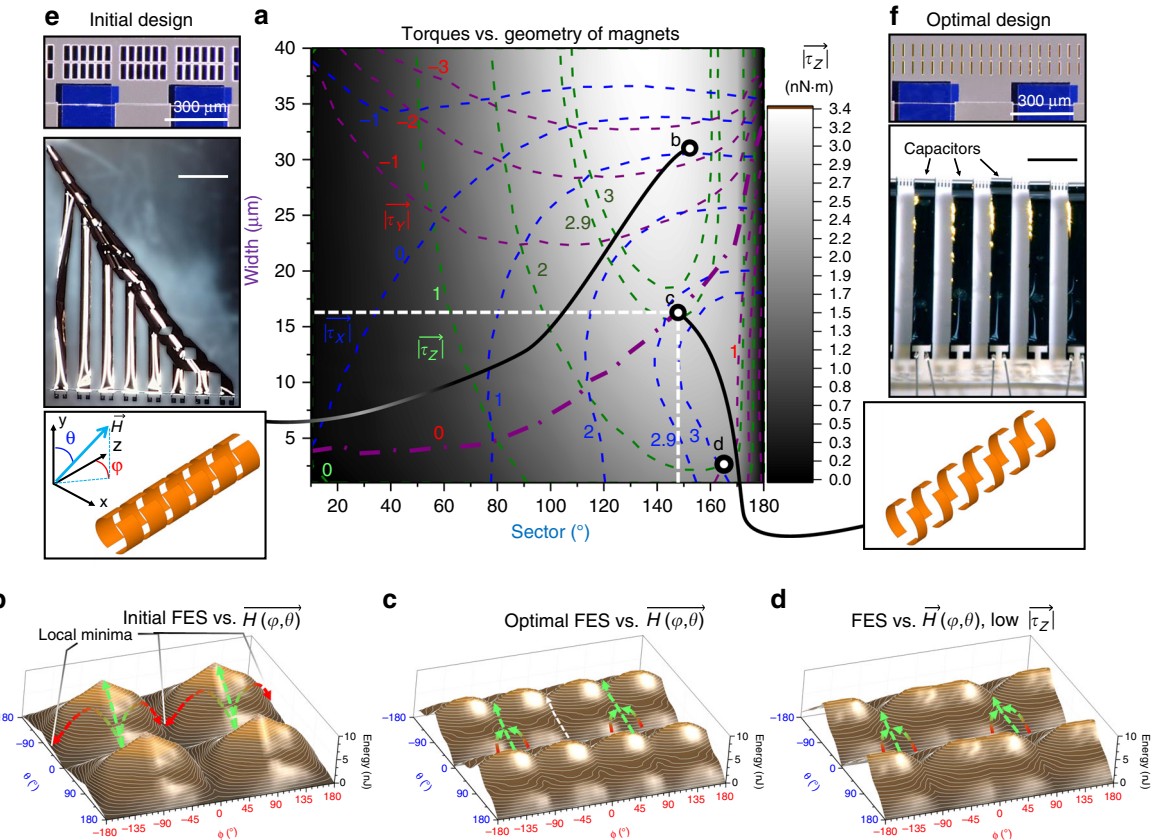

**Fig. 5** Shape-dependent behaviour of magnetic rotor in a rotating magnetic field. **a** Magnetostatic torque map in coordinates of the rotor magnet geometry generating points for the initial design (b), an optimal design (c) of the magnets and a design with good assembly behaviour but low $\overrightarrow{\tau_z}$. **b** Free energy landscapes (free energy surfaces (FES)) of an initial design with characteristic hills and energy minima and potential relaxation pathways of the rotor aligning the structure's z-axis along the field. Green arrows show the pathways the structures have to follow during rotation of the magnet. Red arrows show misaligning relaxation pathways. **c**, **d** FES of best designs with characteristic energy minima and potential relaxation pathways of the rotor aligning the structure's y axis always along the field. For **d**, $\overrightarrow{\tau_y}$ is 30% smaller, which has a direct impact on the maximum rolling length. Green lines on the FES represent rotation pathways for the tube driven by an external motorized magnet. White arrow shows another possible stable position, but it is inaccessible as the initial orientation of the magnetic field is 90°. **e** Misassembly with non-optimized shape of magnetic stripes due to torque that aligns the tube along the field. **f** Magnetic Origami of capacitor array with optimal shape of magnetic stripes where $\overrightarrow{\tau_y}$ aligns the structure perpendicular to the field. **e**, **f** Scale bars are 500 μm

correct orientations of the structure in the magnetic field. However, in this case $\overrightarrow{\tau_z}$ is 30% smaller compared to the design shown in Fig. 5a-II, which leads to a 30% shorter assembly length under the same conditions. Therefore, we consider FES shown in Fig. 5c to be optimal for the assembly of "Swiss-rolls" under a rotating magnetic field.

**Design and verification in a radial magnetic field.** We have realized two layouts containing eight structures attached to a self-assembling rotor containing planar magnet structures and a capacitor stack (Fig. 6a). The fabrication starts with a shapeable polymeric layer stack[21] (SPS) required for self-assembly of the magnetic rotor comprising a 100-nm-thick metal organic SL, 100-nm-thick hydrogel (HG), and 200-nm-thick polyimide (PI). Then the whole structure is covered with 15 nm $Al_2O_3$ and followed by a capacitor stack containing another 15 nm of $Al_2O_3$ sandwiched between two 40 nm Al electrodes. Afterwards, $Co_{90}Fe_{10}$ magnetic layers were deposited and patterned on top of the SPS and the whole structure was passivated with 15 nm of $Al_2O_3$. We finished the planar parallel wafer-scale fabrication process by opening side windows, which are required for the delamination of the layer stack and the assembly process (Fig. 6b).

The structures are then subjected to an etching solution that removes the SL and simultaneously self-assembles the SPS into a rotor tube containing magnetic stripes of the desired geometry (Fig. 6c). Further assembly is driven by a rotating radial magnetic field (Fig. 6d) where the rotor winds the unstrained capacitor stack around itself forming "Swiss-roll" capacitors (Fig. 6d inset). The assembly of the devices with an initial non-optimal design led to entire misassembly with catastrophic damage imposed on structure and performance (Fig. 5e). By contrast, the structures with an optimal design underwent straightforward assembly achieving eight windings (Figs. 5f and 6e). We assembled eight 300-μm wide devices in parallel possessing two different rolling lengths of 5 and 10 mm. The final structures (Fig. 6e) possess outer diameters of about 200 μm with clock spring shape of windings fixed at the substrate from one side and at the rotor from the other. These windings can be further tightened to outer diameters of 100 μm. Diameter tightening is a unique feature of this Origami technique that cannot be achieved in a pure self-assembly approach. The final devices possess an effective footprint of 0.030 mm², which is 50 and 100 times smaller compared to the initial footprint area of the short and long planar structures, respectively.

We characterized these devices in a similar way as in the axial field case and details of this characterization can be found in the

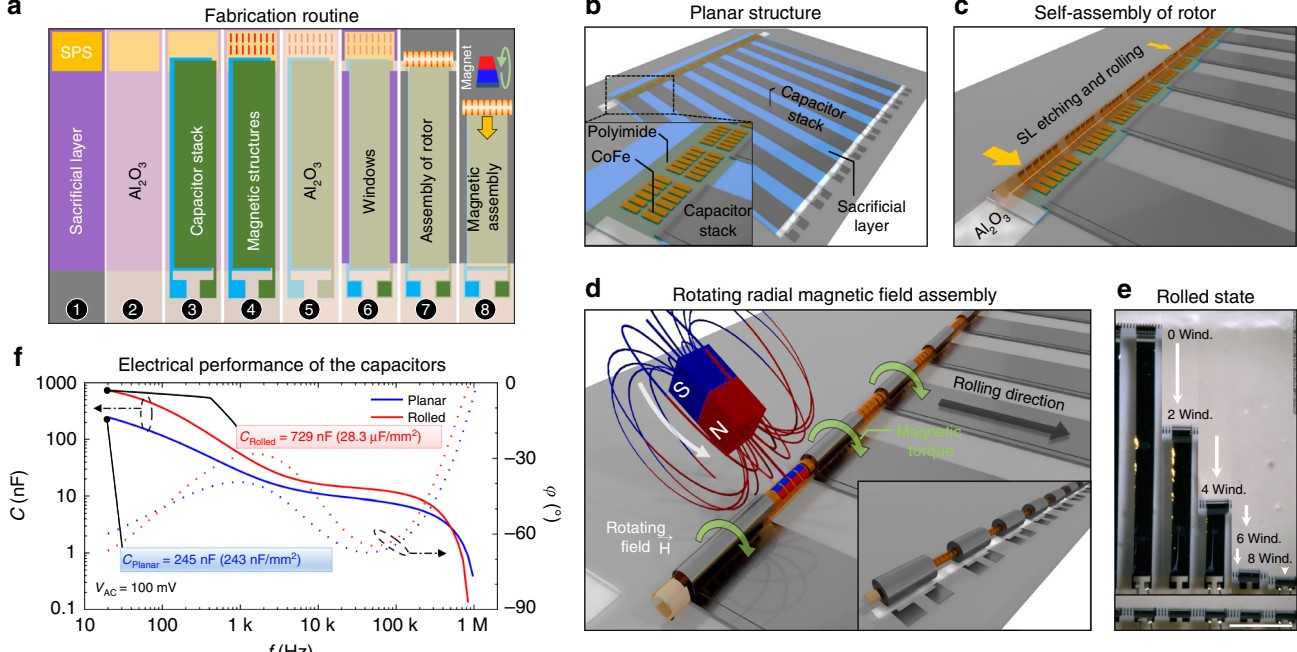

**Fig. 6** Fabrication of capacitors using the radial dynamic magnetic field assembly. **a** Fabrication routine for the radial magnetic field assembling approach.
**b** Planar polymeric and capacitor stack. **c** Magnetic rotor self-assembly during which the sacrificial layer dissolves and the hydrogel layer swells.
**d** Assembly of capacitors in an external rotating magnetic field by winding the unstrained capacitor nanomembrane stack onto the magnetic rotor. (Inset)
Assembled capacitors after fabrication. **e** Time sequence of winding process. Bottom image shows assembled array of capacitors. Scale bar is 1 mm.
**f** Electric performance of planar and assembled capacitors 5 and 10 mm in length measured in $H_2SO_4$ electrolyte

Supplementary Note 6 and Supplementary Fig. 6. This way of assembly allows for an easy solvent exchange, for instance, the rolling solution can be exchanged with an electrolyte. When the devices are placed in an electrolyte (in our case, we use 0.01 M solution of $H_2SO_4$), the capacitance can be increased (Fig. 6f) reaching 245 nF (163 nF/mm²) for planar devices and 729 nF (24.300 nF/mm²) for 5 mm long structures assembled into "Swiss-rolls" 100 μm in diameter. These results reveal an alternative approach demonstrating for the first time aluminum electrolytic capacitors with extremely high capacitance density per footprint area and volumetric efficiency >10⁴ C/m³. Schematics for the radial magnetic field-assisted assembly (Supplementary Fig. 7a) as well as the photo of the actual experimental set-up (Supplementary Fig. 7b) can be found in the supplementary information.

## Discussion

In summary, we have developed a new form of magnetic Origami for parallel wafer-scale fabrication of 3D energy storage devices using magnetic field-assisted assembly of planar nanomembranes. Misassembly issues occurring during mechanical transformation of planar nanomembranes into 3D SRAs have been carefully addressed by simulating and experimental testing of their assembly behaviour in external static and dynamic magnetic fields of two orientations. We have simulated and analysed magnetostatic and mechanical potential energies and plotted corresponding FES, which revealed intricate features such as local minima, potential energy barriers, plateaus, and potential energy slopes that provided deep insights into the dynamics of the assembly process. We could experimentally confirm that these features describe well the spatial assembly behaviour of nanomembranes. We could alter shapes of these surfaces by modifying initial planar structures and applying external static and dynamic magnetic fields to programme the assembly behaviour of SRAs enhancing overall assembly yield to values >90%. 3D assembly of

nanomembranes was achieved over large length scales on the order of centimetres in a reproducible and controlled fashion, unleashing the full potential of magnetic micro-Origami for on-chip manufacturing processes. Magnetic micro-Origami might be a game changer whenever architectures with compact, well-aligned, and multiple rotations of patterned nanomembranes are required, such as 3D photonic crystals[45], cylindrical meta-materials[46], or passive electronic devices, including inductors and transformers.[14,15]

## Methods

**Axial magnetic field approach-related methods.** *Simulation of ferromagnetic helix in axial magnetic field*: Magnetostatic simulation is performed with Maxwell 3D in ANSYS Academics Electromagnetics Suite 17.2 (Ansoft). The fully parameterized simulation model consists of two united user defined primitives "RectHelix" with 36 segments per turn. The helix is modelled using the following parameters: 300 μm width, 60 μm diameter, 15 nm layer thickness, and 30 nm layer spacing. The axial shift is varied from 0 to 15 μm with 0.25 μm increment and the number of windings is varied from 0.25 to 3 with 0.25 increment. The shift is modelled to be equally distributed among the windings. A surrounding box of material "vacuum" with tangential H field boundaries of 796,000 A/m (10 kOe) is used. The material of the helix is set to "nickel" with a value of 1.6 as apparent permeability. The mesh is generated using "Classic" mesh methods. The energy is calculated by integrating the energy density over the helix volume. For each number of windings, the energy at zero shift is defined as zero to determine the relative change of energy dependent on the shift. As the model is symmetric, the calculated energy values are used accordingly for negative shift values from 0 to −15 μm. Forces are calculated as derivative of energy with respect to shift.

*Sacrificial layer*: Methyl cellulose (MC) solution is prepared by mixing 100 mg of methyl cellulose powder (Sigma Aldrich, CAS 9004–67–5) with 100 ml water at 80 °C. The mixture is cooled down to room temperature under continuous stirring. Spin coating of methyl cellulose is performed at 4500 rpm for 35 s on a Si/SiO2 substrate that results in an MC layer of few nanometer thickness. After that, the substrate is baked on a hotplate at 120 °C for 4 min.

*Protection layer*: The MC layer is encapsulated by an $Al_2O_3$ layer deposited by thermal ALD. Deposition is done in a FlexAL system (Oxford Instruments plc, Abingdon, UK) with a chuck temperature of 150 °C using trimethylaluminium (TMA) as precursor at a growth rate of 0.06 nm/cycle.

*Photolithography*: AZ 5214 E photoresist (Microchemicals GmbH, Ulm, Germany) is spincoated at 4500 rpm for 35 s and prebaked on a hotplate at 90 °C

for 2 min. In negative mode, the resist is exposed for 4.5 ms on a µPG501 maskless exposure system (Heidelberg Instruments Mikrotechnik GmbH, Heidelberg, Germany), baked at 120 °C for 2 min and ultraviolet flood exposed to reverse the pattern. This process gives the typical undercut required for lift-off. In positive mode, the resist is exposed for 25 ms. In both negative and positive modes, the resist is developed in AZ 726 MIF developer (Microchemicals GmbH, Ulm, Germany) for 40 s. After developing, a 2-min $O_2$ plasma at 150 W and 0.2 mbar chamber pressure is used to remove any resist residues.

*Electrode layers*: Metal deposition is performed inside an MEB550S electron-beam evaporator (PLASSYS Bestek, Marolles-en-Hurepoix, France) with a base pressure better than $1e^{-7}$ mbar on a water cooled sample holder rotating with 4 rpm. For the capacitor plates, 15 nm Ni is deposited at 0.1 nm/s and 20 nm Cr at 0.3 nm/s. For the contact pads, 5 nm Cr is deposited at 0.1 nm/s for adhesion and 50 nm Au at 0.2 nm/s. Patterning of metal is done by lift-off in acetone from an AZ 5214 E resist mask processed in negative mode.

*Oxide layer*: ALD of the dielectric $Al_2O_3$ layers is done in a FlexAL system with a chuck temperature of 150 °C using a plasma-enhanced process and TMA as precursor at a growth rate of 0.13 nm/cycle.

*Dry etching*: Reactive-ion etching is performed in a Plasmalab 100 tool (Oxford Instruments plc, Abingdon, UK). For etching $Al_2O_3$, the following process parameters are used: 0.06 mbar, 50 sccm CF4, 200 W radio frequency (RF), and 35 °C. The etch rate was measured by a stylus profilometer to be 3–4 nm/min. The process is used to remove $Al_2O_3$ before Cr/Au contact deposition (through the same AZ 5214 E resist mask used for Cr/Au lift-off) as well as to open the windows in $Al_2O_3$ and expose the sacrificial layer at the starting areas (through an AZ 5214 E resist mask processed in positive mode).

*Assembling process*: The assembly is performed inside a dipole electromagnet with 50 mm pole gap and 115 mm pole face diameter at 10 kOe magnetic field. The sample is fixed inside a PTFE holder with vacuum suction. Further details can be found in Supplementary Note 4.

**Radial magnetic field-related methods.** *Simulation of ferromagnetic stripes in radial magnetic field*: For the second approach, we used tubular-shaped magnetic stripes (Fig. 3d, e bottom). For the simulation, we used CoFe as magnetic material with a saturation field of 1591 A/m (20 kOe). An *H* field of 80 kA/m (1000 Oe) was applied in calculation of magnetic torques, which is comparable to the maximum magnetic field measured in the rolling set-up at the sample level. Torques generated around *x*, *y*, and *z* axes were calculated for various orientations of the magnetic field vector around the structure. Geometry parameters of the stripes, the sector and the stripe width, were parametrically swept in the range of 10–180° and 2–40 µm, respectively.

*Processing SPS*: The glass cleaning and synthesis process of polymeric materials was described in our previous works.[12,14,47,48] For this work, SL, HG, and PI polymeric materials were diluted to the point where they give a maximum thickness in the range of 100, 100, and 200 nm, respectively, after the spincoating process. The polymeric stack consisting of SL, HG, and PI was formed in a sequential photolithographic process. We applied 3 ml of each solution on glass substrates through 1-µm glass fibre filters. Then the substrate was spun at 500 rpm with the acceleration of 50 rpm/s for 10 s for spreading of the polymer solution across the substrates surface. Then the sample was accelerated with the rate of 500 rpm/s to the speed of 3000 rpm, at which it was kept for 30 s. SL, HG, and PI layers were dried after the spin coating process at 35 °C for 5 min, 40 °C for 5 min, and 50 °C for 10 min, respectively. Finally, samples were exposed through a photomask by a SUSS MA6 (1000 W) mask aligner, developed, and post baked following the process described in our previous works.[12,14,47,48]

*Electrode layer*: The 40 nm aluminium metal layer was deposited by an MEB550S electron-beam evaporator (PLASSYS Bestek, Marolles-en-Hurepoix, France). The base pressure was better than $1e^{-7}$ mbar. The deposition tilting angle and rate were 10° and 4 Å/s, respectively. The sample was rotating at 6 rpm/min. Then this layer was patterned using AZ5214E photoresist (Microchemicals GmbH, Ulm, Germany) and AZ MIF726 (Microchemicals GmbH, Ulm, Germany) developer.

*Magnetic layer*: CoFe magnetic layer stripes were made via magnetron sputter deposition in a high-vacuum chamber with a base pressure of $4 \times 10^{-7}$ mbar. To introduce stress in the CoFe layer, magnetic material was deposited in two steps. First layer was done at Ar sputter pressure of $9 \times 10^{-4}$ mbar that resulted in layer thickness of 160 nm. Second layer was deposited at Ar sputter pressure of $5 \times 10^{-3}$ mbar that resulted in layer thickness of 240 nm. Sputtering power was kept at 100 W for both layers.

*Oxide layer*: $Al_2O_3$ insulation layers is done via a PE-ALD process with a chuck temperature of 220 °C and TMA as precursor at a growth rate of 0.13 nm/cycle.

*Dry etching*: Al and $Al_2O_3$ was selectively etched in the Plasmalab 100 tool (Oxford Instruments plc, Abingdon, UK). Etching gas composition and working process were obtained from elsewhere[49]. In brief, this process was performed at 0 °C over 8 min. The pressure was kept at 15 mbar. RF and inductively coupled plasma powers were kept at 40 W and 300 W, respectively. The used gas mixture was composed of Cl, BCl3, and Argon supplied in a ratio of 2:6:14, respectively, and the total flow of 22 ml/min. Helium backside flow was set at 10 ml/min. For this process, we used a standard photoresist pattern on top of the Al layer.

*Rolling process of polymeric structures*: Self-assembly of rotors was performed by a selective etching of SL layer and a simultaneous swelling of the HG in a solution of strongly chelating agent. This solution was prepared from 10 g of sodium-ethylenediaminetetraacetic acid (Alfa Aesar, UK) dissolved in 500 ml of deionized water.

*Magnetic assembly process*: Samples were assembled using the in-house built set-up (Supplementary Fig. 7). The set-up consists of a controlling unit-based (Arduino XNUCLEO STM32F103 board Waveshare Electronics, Shenzhen, China), micro-stepping module (DRV8825 and NEMA13 step motors Eckstein GmbH, Clausthal-Zellerfeld, Deutschland), and a linear motion stage. Assembly of nanomembranes is assisted by a rotating magnetic field of a permanent magnet attached to a stepper motor (Supplementary Fig. 7a). The sample is placed on the rolling set-up (Supplementary Fig. 7b) above the magnet with a constant field of 80 kA/m (1 kOe) magnetic field. The magnet position was automatically shifted keeping a constant distance to the sample.

**Electrical characterization.** Electrical characterization was performed in a probe station (PM5 and Summit 12000 provided by Cascade Microtech Inc., Thiendorf, Germany) using a precision LCR Meter (E4980A Agilent Technologies Inc., Santa Clara, USA) at a constant voltage of 100 mV. The current–voltage characteristics were measured using a source-meter (2614B Keithley Instruments, Solon, USA).

## Data availability

The authors declare that all data supporting the findings of this study are available within the paper and its supplementary information files.

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

## Acknowledgements

The support of the clean room team headed by R. Engelhard (IIN/IFW Dresden) and the assistance in development of the experimental set-ups by the research technology department of IFW Dresden is greatly appreciated. We are grateful for fruitful discussions with D. Grimm and S. Harazim from IFW Dresden as well as A. Ando and S. Suzuki from Murata Manufacturing Co., Ltd., especially at the beginning of this project. We further thank C. Krien and I. Fiering (IIN/IFW Dresden) for the deposition of metallic thin films, S. Baunack (IIN/IFW Dresden) for FIB/SEM analysis as well as Ulrike Nitzsche (ITF/IFW Dresden) for the help with the calculation server. This work was supported by the German Research Foundation DFG (Gottfried Wilhelm Leibniz Program and KA5051/1–1).

## Author contributions

O.G.S. and D.K. conceived the idea. F.G. designed and performed experiments for axial field-assisted assembly and D.D.K. designed and performed experiments for radial field-assisted assembly. D.K. wrote the manuscript with input from all authors and supported experimental design. O.G.S. and D.K. supervised the work. All authors participated in data analysis and discussion.

## Additional information

**Competing interests:** The authors declare no competing interests.

