## [Peer Review File · Nature Communications]

Reviewers' comments:

Reviewer #1 (Remarks to the Author):

The group has demonstrated a number of interesting things using nanomembranes. The applications that they addressed are very unique and many of them are even very surprising to me. In this manuscript, they introduced a magnetic field assisted assembly of 3D nanomembrane architectures. The work is very interesting and intriguing. Specifically, by applying static and dynamic magnetic fields the authors demonstrated "Swiss-roll" electrostatic capacitors with high length-to-width aspect ratio nanomembrane. 3D assembly of nanomembranes was achieved over large length scales on the order of centimeters in a controlled fashion. Significantly enhanced capacitance density due to reduced footprint area and improved ultrahigh volumetric efficiency is achieved.

The experiment was clearly described and the data are systematically analyzed. I did not see any issues in the manuscript. The manuscript in its present form can be published in the journal.

As an option to the authors, I would appreciate the authors could comment on the following two questions:

- 1) In terms of the thickness choice of the dielectric layer here as Al₂O₃, other than the consideration of the break down voltage, leakage current and rolling speed, would the choice of the dielectric layer affect the optimal planar design parameters such as shapes and geometry?
- 2) Could the authors comment on the potential applications in other areas beyond capacitors using the magnetic field assisted assembly approach? Is the method restricted for use in other applications due to the requirement of uniform distribution ferromagnetic material and therefore its response to an external magnetic field?

Overall, the work is excellent and should be accepted.

Zhenqiang Ma

Reviewer #2 (Remarks to the Author):

During the last years, the group of O. Schmidt demonstrated a lot of skill in the "rolling-up" of thin films and nanomembranes into nano-cylinders. Although such rolled-up objects are not really compatible with the planar technology commonly used for the fabrication of integrated circuits, the Schmidt group brought this approach to perfection, and they were able to demonstrate the fabrication and testing of several functional nanodevices.

The current paper is along this line. The authors first present a new concept to manipulate and align the rolled-up nanomembranes with magnetic fields. In some detail and using a somewhat uncommon writing style that makes use of colored and all-capital-letter words in the text, they explain how their so-called "magnetic origami" works. Then they describe the fabrication of rolled capacitors and the characterization of the frequency behavior of the capacitance. The manuscript ends with a comparison to commercial capacitors, some speculation about the future usage of rolled-up capacitors and the statement that the described capacitors are "most lightweight and compact ... with unrivaled performance". Here, the authors maybe have overlooked that the field of nanocapacitors is quite active and that for example Zhang et al. (ACS Nano 12, 10301, 2018) have reported the fabrication of significantly lighter all-carbon-capacitors that show similar performances of up to 0.5 $\mu\text{F}/\text{cm}^2$. Of course, Zhang's capacitors are not rolled-up but his work - and others- should be correctly cited and discussed in the manuscript. Once, these revisions are made, I would recommend the publication.

Reviewer #1 (Remarks to the Author):

The group has demonstrated a number of interesting things using nanomembranes. The applications that they addressed are very unique and many of them are even very surprising to me. In this manuscript, they introduced a magnetic field assisted assembly of 3D nanomembrane architectures. The work is very interesting and intriguing. Specifically, by applying static and dynamic magnetic fields the authors demonstrated “Swiss-roll” electrostatic capacitors with high length-to-width aspect ratio nanomembrane. 3D assembly of nanomembranes was achieved over large length scales on the order of centimeters in a controlled fashion. Significantly enhanced capacitance density due to reduced footprint area and improved ultrahigh volumetric efficiency is achieved.

The experiment was clearly described and the data are systematically analyzed. I did not see any issues in the manuscript. The manuscript in its present form can be published in the journal.

We thank Reviewer #1 for his positive remarks.

As an option to the authors, I would appreciate the authors could comment on the following two questions:

1) In terms of the thickness choice of the dielectric layer here as Al₂O₃, other than the consideration of the break down voltage, leakage current and rolling speed, would the choice of the dielectric layer affect the optimal planar design parameters such as shapes and geometry?

We thank the reviewer for the question. The choice of the material was made considering electrical properties of ALD Al₂O₃. The mechanical properties such as the elastic modulus of the material will have an impact on the final diameter of the Swiss-roll architecture, but will not critically affect the design of the planar structure.

2) Could the authors comment on the potential applications in other areas beyond capacitors using the magnetic field assisted assembly approach? Is the method restricted for use in other applications due to the requirement of uniform

distribution ferromagnetic material and therefore its response to an external magnetic field?

We thank the reviewer for the question. Well-aligned multiple-rotations of patterned nanomembranes are required, for instance when constructing delicate 3D photonic crystals [Phys. Rev. A. 87, 041803 (2013)] or cylindrical metamaterials [Nano Lett. 10, 1 (2010)]. More obvious applications are other passive devices such as efficient inductor and transformer components, the reports on which were already included in the reference list [14, 15]. We added appropriate text to the conclusion part of the manuscript.

Overall, the work is excellent and should be accepted.

Thank you for this final remark.

Reviewer #2 (Remarks to the Author):

During the last years, the group of O. Schmidt demonstrated a lot of skill in the “rolling-up” of thin films and nanomembranes into nano-cylinders. Although such rolled-up objects are not really compatible with the planar technology commonly used for the fabrication of integrated circuits, the Schmidt group brought this approach to perfection, and they were able to demonstrate the fabrication and testing of several functional nanodevices.

We thank the Reviewer for these positive remarks.

The current paper is along this line. The authors first present a new concept to manipulate and align the rolled-up nanomembranes with magnetic fields. In some detail and using a somewhat uncommon writing style that makes use of colored and all-capital-letter words in the text, they explain how their so-called “magnetic origami” works. Then they describe the fabrication of rolled capacitors and the characterization of the frequency behavior of the capacitance. The manuscript ends with a comparison to commercial capacitors, some speculation about the future usage of rolled-up capacitors and the statement that the described capacitors are “most lightweight and compact ... with unrivaled performance”. Here, the authors

maybe have overlooked that the field of nanocapacitors is quite active and that for example Zhang et al. (ACS Nano 12, 10301, 2018) have reported the fabrication of significantly lighter all-carbon-capacitors that show similar performances of up to $0.5 \mu\text{F}/\text{cm}^2$.

Of course, Zhang's capacitors are not rolled-up but his work -and others- should be correctly cited and discussed in the manuscript. Once, these revisions are made, I would recommend the publication.

We thank the reviewer for these kind comments. However and with due respect, we point out that our capacitors have capacitance densities up to $150 \mu\text{F}/\text{cm}^2$ (Fig. 4e), in other words 300 times larger than the one mentioned by the Reviewer. Still, it will be interesting to incorporate 2D, graphene layers or other all carbon materials into the "Swiss-roll" microcapacitors to further enhance their performance. We have included appropriate text in the manuscript, also citing the publication mentioned by the reviewer.

REVIEWERS' COMMENTS:

Reviewer #1 (Remarks to the Author):

The authors have addressed the comments properly and have improved the quality of the manuscript. The manuscript is acceptable.

Reviewer #2 (Remarks to the Author):

OK !